# Sponsorship of national and regional professional paediatrics associations by companies that make breast-milk substitutes: evidence from a review of official websites

Laurence M Grummer-Strawn,[1] Faire Holliday,[2] Katharina Tabea Jungo,[3] Nigel Rollins[4]

For numbered affiliations see end of article.

**Correspondence to**
Dr Laurence M Grummer-Strawn;
grummerstrawnl@who.int

## ABSTRACT

**Objectives** Professional paediatrics associations play an important role in promoting the highest standard of care for women and children. Education and guidelines must be made in the best interests of patients. Given the importance of breastfeeding for the health, development and survival of infants, children and mothers, paediatric associations have a particular responsibility to avoid conflicts of interest with companies that manufacture breast-milk substitutes (BMSs). The objective of this study was to investigate the extent to which national and regional paediatric associations are sponsored by BMS companies.

**Methods** Data were collected on national paediatric associations based on online searches of websites and Facebook pages. Sites were examined for evidence of financial sponsorship by the BMS industry, including funding of journals, newsletters or other publications, conferences and events, scholarships, fellowship, grants and awards. Payment for services, such as exhibitor space at conferences or events and paid advertisements in publications, was also noted.

**Results** Overall, 68 (60%) of the 114 paediatric associations with a website or Facebook account documented receiving financial support from BMS companies. Sponsorship, particularly of conferences or other events, was the most common type of financial support. The prevalence of conference sponsorship is highest in Europe and the Americas, where about half of the associations have BMS company-sponsored conferences. Thirty-one associations (27%) indicated that they received funding from BMS companies as payment for advertisements or exhibitor space. Only 18 associations (16%) have conflict of interest policies, guidelines, or criteria posted online.

**Conclusion** Despite the well-documented importance of breastfeeding and the widespread recognition that commercial influences can shape the behaviours of healthcare professionals, national and regional paediatric associations commonly accept funding from companies that manufacture and distribute BMS. Paediatric associations should function without the influence of commercial interests.

### Strengths and limitations of this study

► This is the first study to systematically document the extent of conflicts of interest in national paediatric associations with regard to manufacturers of breast-milk substitutes.
► Data from websites were available for 75% of national paediatric associations.
► Data were objectively collected and analysed based on online documentation.
► Funding that has not been documented on websites was not captured, potentially leading to an underestimate of the extent of industry funding.
► This study was not able to capture information on the amount of funding received or on how it compares to funding from other sources.

## INTRODUCTION/BACKGROUND

Breastfeeding is critical for the health, development and survival of infants, children and mothers. Recent analyses have concluded that an estimated 820 000 infant and child deaths still occur each year from substandard breastfeeding practices or non-breastfeeding.[1] Nearly half of diarrhoea episodes and one-third of respiratory infections are due to inadequate breastfeeding practices. Longer breastfeeding is associated with a 13% reduction in the likelihood of overweight and obesity and a 35% reduction in the incidence of type 2 diabetes. An estimated 20 000 maternal deaths from breast cancer could be prevented each year by improving breastfeeding. Economically, increasing rates of breastfeeding could add US$ 300 billion to the global economy annually by helping to foster smarter, more productive workers and leaders.[2]

Marketing of breast-milk substitutes (BMSs) has effectively reduced rates of breastfeeding.

Globally, sales of BMSs are growing almost eight times as quickly as the world's population.[3] It is estimated that the six largest BMS manufacturers spend over US$7 billion annually on marketing activities.[4] A significant portion of this marketing is targeted at healthcare providers and institutions. Given the importance of breastfeeding, health professional advice and support for breastfeeding should not be influenced by the commercial interests of BMS manufacturers.

The widespread involvement of BMS companies in healthcare has long been recognised and documented.[5–7] Recent studies in Mexico, Chile and Ecuador have reported extensive use of BMS promotional materials in health facilities, distribution of free formula samples, gifts to health workers, donations of equipment and sponsorship of events.[8–10] In Bangladesh, many mothers are advised by healthcare workers, especially in private clinics, to use BMS without any prior counselling on breastfeeding.[4] Health workers in China were paid by a BMS company to recommend infant formula to new or expectant mothers, leading to an US$12 million settlement against the company.[11]

The International Code of Marketing of Breast-milk Substitutes, adopted by the World Health Assembly in 1981, delineates a number of steps to limit the ways in which industry uses healthcare workers to promote its products.[12] In 2005, the World Health Assembly urged countries to 'ensure that financial support and other incentives for programmes and health professionals working in infant and young child health do not create conflicts of interest'.[13] The 69th World Health Assembly (2016) passed a resolution on WHO's 'Guidance on ending inappropriate promotion of foods for infants and young children', calling on health professionals to 'fulfil their essential role in providing parents and other caregivers with information and support on optimal infant and young child feeding practices and to implement the guidance recommendations' (paragraph 4).[14] The guidance recommendations[15] state that health professional associations should not 'accept equipment or services from companies that market foods for infants and young children; accept gifts or incentives from such companies…[or] allow such companies to sponsor meetings of health professionals and scientific meetings' (paragraph 17). (The Implementation Manual[16] for the Guidance provides more information).

Although anecdotal information exists about the relationship between the BMS industry and paediatric associations, it has not yet been examined in a systematic way. The objective of this study was to investigate the extent to which national and regional paediatric associations are sponsored by BMS companies.

## METHODS

Data were collected on national (n=146) or regional (n=6) paediatric associations listed on the webpage of the International Paediatric Association.[17]

Online searches were conducted for evidence of funding of paediatric associations by the BMS industry. This involved first determining which associations had official websites or, in the absence of official websites, Facebook accounts. Searches were then conducted within the online presence for these associations. For associations with a website, searches were conducted on the main website, as well as any associated journal websites, conference websites, and charity or non-profit branch or foundation websites, as available.

In addition, the Facebook accounts of 10 randomly selected associations that also had official websites were searched. As this process did not render any information in addition to that which was found on the website itself, no additional Facebook accounts were searched.

**Table 1** Type of funding from breast-milk substitute (BMS) companies identified by online searches

| Category | Criteria |
| --- | --- |
| *Sponsorship* | |
| Funding of journals, newsletters or other publications | ▶ Written indication of BMS companies as sponsors of the journal, magazine, or newsletter |
| Funding for conferences and events | ▶ BMS company logos on conference webpage<br>▶ Written indication of BMS companies as sponsors<br>▶ Written indication of satellite symposia or other events sponsored by BMS companies |
| Funding for scholarships, fellowship, grants and awards | ▶ Written indication of BMS companies as sponsors of scholarships, awards, grants, or fellowship funding |
| Funding for websites or general use | ▶ BMS company logos on home page or sponsors/partners page<br>▶ Written indication of BMS companies as sponsors<br>▶ Written promotion of a particular BMS company |
| *Payment for services* | |
| Paid advertisements in publications | ▶ Advertisements of BMS companies in online publications |
| Exhibitor space at conferences or events | ▶ Written indication of BMS companies as exhibitors<br>▶ Photos of BMS exhibitions at conferences |
| Associations' conflict of interest policies, guidelines or criteria | ▶ Existence of any official document of the association that mentions conflicts of interest (includes official conflict of interest documents, codes of ethics, association statutes, etc.) |

Data were collected by two research assistants in June–August 2017. Reliability was assessed by having both research assistants conduct searches for the same 10 member associations and compare results. Minor discrepancies were encountered as to where on the website certain information was encountered, but in no case was the overall assessment of receipt of funding from BMS companies different. After this process, the remaining associations were divided and assigned to each research assistant. Some websites were in a language that was familiar to one research assistant or the other and were therefore assigned accordingly. Websites in languages other than English, French, German, Italian, Portuguese or Spanish were translated using online translators and additional interpretation was sometimes provided by other WHO staff or interns. The rest were divided randomly.

The websites were examined for any logos or names of BMS companies found in acknowledgments, funding sources, advertisements and lists of exhibitors or sponsors of conferences. Where funders or sponsors were unknown, additional web searches were conducted to fully understand the nature of the donor. Home pages and all sub-pages, PowerPoint presentations, event photos and online documents, such as pamphlets, education sheets, newsletters and publicly accessible journals were included in the search. Content that required a membership to access, such as subscription journals, were not reviewed.

Searches were also conducted on the websites of BMS companies. Due to the high number of such companies (and national affiliates), it was impossible to identify or search the websites of all. In order to be systematic, the websites of 'infant formula' companies listed on the Baby Milk Action website were searched.[18]

Information was collected on the purpose of funding (table 1) and classified as either 'sponsorship' or 'payment for services'. Funding was considered to be 'sponsorship' if there appeared to be no specific services provided to the donor other than acknowledgement. Funding was considered to be 'payment for services' if the paediatric association provided direct benefits to the company, such as through advertisements in a publication or exhibition space at a conference.

Data were entered into a spreadsheet and checked by both research assistants to verify the relevance of the information. For each category of funding, 'sponsorship' or 'payment for services', the association was counted as 'yes' if there was evidence of financial contribution from BMS companies in that category. In addition to funding, information was collected on whether the association had policies or criteria on conflicts of interest available online.

### Patient and public involvement
No patients were involved in the study.

## RESULTS
Of the 152 paediatric associations, websites were identified for 109 and an additional five had Facebook pages, yielding a total of 114 associations with an online presence (75%). Overall, 68 of the paediatric associations with an online presence (60%) documented receipt of some kind of financial support from BMS companies, either in the form of sponsorship or payment for services (table 2). This was highest in the Americas region, with 23 of 28 (82%) associations receiving some sort of funding from BMS companies. In Europe, 21 of 32 (66%) associations received BMS funding, while in Asia and Africa it was 15 of 30 (50%) and 8 of 21 (38%), respectively. One association in Oceania was found to have received funding from BMS companies. However, there are only three paediatric associations identified in this region, and the results should be considered accordingly.

**Table 2** Number of paediatric associations that receive financial support* from manufacturers of breast-milk substitutes based on website review, by type of support

| | World (n=114) | Africa (n=21) | Americas (n=28) | Asia (n=30) | Europe (n=32) | Oceania (n=3) |
|---|---|---|---|---|---|---|
| *Any financial support n, (%)* | 68 (60%) | 8 (38%) | 23 (82%) | 15 (50%) | 21 (66%) | 1 (33%) |
| *Sponsorship n, (%)* | 60 (53%) | 8 (38%) | 20 (71%) | 13 (43%) | 19 (59%) | 0 (0%) |
| Funding of journals, newsletters or other publications | 4 (4%) | 0 (0%) | 1 (4%) | 3 (10%) | 0 (0%) | 0 (0%) |
| Funding for conferences and events | 43 (38%) | 7 (33%) | 13 (46%) | 7 (23%) | 16 (50%) | 0 (0%) |
| Funding for scholarships, fellowship, grants and awards | 10 (9%) | 0 (0%) | 5 (18%) | 3 (10%) | 2 (6%) | 0 (0%) |
| Funding for websites or general use | 15 (13%) | 2 (10%) | 5 (18%) | 3 (10%) | 5 (16%) | 0 (0%) |
| Purpose of funding not stated | 16 (14%) | 5 (24%) | 7 (25%) | 1 (3%) | 3 (9%) | 0 (0%) |
| *Payment for services n, (%)* | 31 (27%) | 1 (5%) | 14 (50%) | 5 (17%) | 10 (31%) | 1 (33%) |
| Paid advertisements in publications | 14 (12%) | 1 (5%) | 5 (18%) | 3 (10%) | 5 (16%) | 0 (0%) |
| Exhibitor space at conferences or events | 22 (19%) | 1 (5%) | 10 (36%) | 2 (7%) | 8 (25%) | 1 (33%) |

*Financial support includes sponsorship, for which no specific services are provided to the donor other than acknowledgement, and payment for services.

Sponsorship was the most common type of funding received by paediatric associations. Overall, 60 of 114 (53%) websites of paediatric associations indicated sponsorship by BMS companies. Forty-three associations (38%) have conferences or other events sponsored by BMS companies. The prevalence of conference sponsorship is highest in Europe and the Americas, where about half of the associations have BMS company-sponsored conferences.

In addition to conferences, paediatric associations may receive other types of sponsorship from BMS companies, though less frequently. Fifteen associations (13%) receive general sponsorship of the association or its website, 10 associations (9%) receive sponsorship for scholarships, awards, grants or fellowships and 4 (4%) have publications that are sponsored.

Thirty-one associations (27%) indicated that they received funding from BMS companies as payment for services (eg, journal advertisements or exhibition space at conferences). This was highest in the Americas, with 14 of 28 (50%), and in Europe, with 10 of 21 (31%). Fourteen (12%) were found to have BMS company advertising in their online publications (journals, magazines or newsletters). Worldwide, only one-fifth (22 of 114) of the associations had exhibitions by the BMS industry at conferences.

Documentation of association sponsorship was also found on BMS company websites, although most of this information was captured on the association websites. While 16 associations were identified as recipients of funding on BMS company websites, only 3 of these associations did not already have documentation of this on their own website. In general, it was not possible to determine the purpose of the sponsorship on the BMS company websites.

Only 18 (16%) associations published conflict of interest policies, guidelines or criteria. Many of these address conflicts of interest among individuals in leadership positions or the need to declare interests when making presentations. Policies on sponsorship or funding typically gave general criteria that donors cannot compromise the vision, mission or values of the association but did not specify how this would be determined. Two associations listed specific industries that they would not work with (eg, tobacco and arms), but in neither case were BMS manufacturers on that list.

Associations that receive financial sponsorship from BMS companies are more likely to have a conflict of interest policy (13 of 60 associations, or 22%) than those that are not sponsored (5 of 54 associations, or 9%).

## DISCUSSION

Despite the well-documented importance of breastfeeding and the widespread recognition of how commercial influences shape the behaviours of healthcare professionals, national and regional paediatric associations commonly accept funding from companies that make BMS. This study found that 53% of association websites acknowledge receiving sponsorship from BMS companies. In addition, when payment for advertisements or exhibitor space is included, it can be seen that 60% of associations receive financial support from BMS companies.

A conflict of interest occurs when a set of conditions in which professional judgement concerning a primary interest (such as a patient's welfare or the validity of research) tends to be unduly influenced by a secondary interest (such as financial gain).[19] Rodwin[20] defined it as when an individual has an obligation to serve a party or perform a role and the individual has either incentives or conflicting loyalties which encourage the individual to act in ways that breach his/her obligations.

Each type of sponsorship presents its own challenges and has the potential to create a conflict of interest. Perhaps of greatest concern is the widespread sponsorship of conferences and other events at which paediatricians meet and disseminate research. The impact of the BMS industry at these events—either as direct sponsors, sponsors of symposia or presenters of information—is not to be underestimated. For example, as Fabbri et al[21] noted, funding of conferences and satellite symposia may bias the scientific content presented at such events.

Furthermore, accepting any kind of support from the industry 'creates a sense of obligation and loyalty to the company in question'.[22] Even receiving foods and beverages sponsored by industry at these events or receiving conference materials bearing company logos may cause physicians to feel a subconscious onus to reciprocate.[23] These factors have the potential to influence what physicians prescribe to their patients.[21 23] Exhibitions by the BMS industry at conferences are likewise concerning, particularly as participants are often required to walk through the exhibit hall to access scientific events.

Industry sponsorship of medical journals is concerning in that it may shape the content that is presented to health professionals. Funding for scholarships and grants has the potential to impact what topics are researched, thus influencing the field for years to come.[24] Recommendations on clinical practice may be unduly influenced by close relationships between professional expert bodies and the BMS industry. For example, van Tulleken[25] has noted that prescriptions of specialist formula milks for cow's milk protein allergy have increased dramatically over the past decade based on the guidelines of several expert groups. Ten of the 12 authors of the 2012 European Society for Paediatric Gastroenterology, Hepatology, and Nutrition guidelines on diagnosis and management of cow's-milk protein allergy[26] and all authors of the international Milk Allergy in Primary Care (iMAP) guideline on cow's milk allergy[27] declared financial interests with infant formula manufacturers.

This study was unable to document the actual amount of funding provided by the BMS companies, as this information is rarely posted on public websites. Professional medical associations are not required to share their financial records.[28] Dalsing[24] has estimated that professional medical associations receive 30%–50% of their budgets

from industry relationships. Many associations are currently dependent on funding from BMS companies for operating expenses.

Although refusing sponsorship from the BMS industry may reduce paediatric associations' budgets, there are alternatives that could lessen the impact of the financial relationships. Schofferman et al[28] suggests that associations raise their dues, increase recruitment or downsize some of their more expensive activities. It has been estimated that the American Academy of Pediatrics could raise its dues US$50 to cover the cost of refusing BMS sponsorship.[29]

Only 18 associations (16%) had posted online some sort of policy to manage conflicts of interest. Interestingly, associations with a conflict of interest policy are actually more likely to accept sponsorship from BMS companies than those that do not have such policies. This finding is consistent with previous research by Fabbri et al,[21] showing that Italian professional medical associations with a conflict of interest policy were no less likely to have sponsorship from industry as those without. It may be the case that conflict of interest policies actually make it easier to accept funding or it may be that once a decision has been made to accept funding, the association sees the need to write down a policy to justify the acceptance and to govern how the funds will be used.

### Strengths

This is the first time that the sponsorship of national and regional paediatric associations by BMS companies has been documented in a systematic way. Out of 152 known associations, we found online information about 114 of them, allowing for a regional breakdown of patterns. Data were objectively collected and analysed. We were able to document the purpose of the funding received and examine the existence and content of policies about funding and conflicts of interest.

### Limitations

As the research was limited to associations' online presence, some information was likely missed. The associations were not contacted directly to confirm the completeness of the website documentation. No information was available for the 38 associations with neither a website nor Facebook page, although these associations were generally in small countries and may be quite small associations. Review of websites not in English, French, German, Italian, Portuguese or Spanish may not have been entirely complete. Not all funding may be acknowledged on websites. It was outside of the scope of this project to investigate print versions of publications, which may be more likely to display advertisements than online versions of journals. As a result, the extent of industry funding is likely underestimated.

This study was not able to capture information on the amount of funding received or on how it compares to funding from other sources.

### CONCLUSION

This study has documented that paediatric associations regularly receive funding from BMS companies, particularly through sponsorship of conferences and meetings, as well as publications, scholarships, fellowship, grants and awards. Paediatric associations are tasked above all else with safeguarding the health of infants, children and mothers and promoting the highest standard of care, including the protection, promotion and support of breastfeeding. WHO recommends that this should be without influence from industry. Policies on conflicts of interest are relatively rare and do not appear to limit the decision to accept funds from the BMS industry. In accordance with the World Health Assembly Resolution 69.9, paediatric associations should refuse sponsorship from the BMS industry and identify alternative funding models especially with respect to the management and style of conferences.

**Author affiliations**
[1]Department of Nutrition for Health and Development, World Health Organization, Geneva, Switzerland
[2]Global Health & Health Disparities, Colorado School of Public Health, Ft. Collins, Colorado, USA
[3]Institute of Primary Health Care (BIHAM), University of Bern, Bern, Switzerland
[4]Department of Maternal, Newborn, Child and Adolescent Health, World Health Organization, Geneva, Switzerland

**Contributors** LMG-S conceived the project, drafted the overall paper and tables and is responsible for the overall content as guarantor. FH oversaw and conducted the data collection and managed the data repository. KTJ participated in the data collection, conducted the data analyses and contributed part of the text. NR contributed to the project design and suggested significant revisions to the paper. All four authors reviewed the final revision.

**Funding** All authors are staff or interns at the World Health Organization. No separate funding was obtained for this study.

**Disclaimer** The authors alone are responsible for the views expressed in this article and they do not necessarily represent the decisions, policy or views of the institutions with which they are affiliated.

**Competing interests** None declared.

**Patient consent for publication** Not required.

**Provenance and peer review** Not commissioned; externally peer reviewed.

**Data sharing statement** All data relevant to the study are included in the article or uploaded as supplementary information.

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
