## [Reviewer comments · BMJ Open]

ARTICLE DETAILS

TITLE (PROVISIONAL)	Sponsorship of National and Regional Professional Paediatrics Associations by Companies that Make Breast-milk Substitutes: Evidence from a Review of Official Websites
AUTHORS	Grummer-Strawn, Laurence; Holliday, Faire; Jungo, Katharina; Rollins, Nigel

VERSION 1 – REVIEW

REVIEWER	Joel Lexchin Professor Emeritus York University Canada In 2015-2018, Joel Lexchin was a paid consultant on three projects: one looking at indication-based prescribing (United States Agency for Healthcare Research and Quality), a second to develop principles for conservative diagnosis (Gordon and Betty Moore Foundation) and a third deciding what drugs should be provided free of charge by general practitioners (Government of Canada, Ontario Supporting Patient Oriented Research Support Unit and the St Michael's Hospital Foundation). He also received payment for being on a panel that discussed a pharmacare plan for Canada (Canadian Institute, a for-profit organization), a panel at the American Diabetes Association, for a talk at the Toronto Reference Library and for writing a brief for a law firm. He is currently a member of research groups that are receiving money from the Canadian Institutes of Health Research and the Australian National Health and Medical Research Council. He is member of the Foundation Board of Health Action International and the Board of Canadian Doctors for Medicare.
REVIEW RETURNED	16-Jan-2019

GENERAL COMMENTS	This study is a systematic look at the issue of where medical societies get their funding and documents the extent to which paediatric associations receive funding from companies that make breast milk substitutes. It is important to highlight conflicts of interest between medical societies and commercial interests for the reasons that the authors outline. 1. Page 6, line 55: There should be a citation included to support the statement about the way that BMS companies are involved in health care.2. Page 12, lines 32-34: The authors should make it clear that the 30-50% figure is Dalsing's personal estimate and that he doesn't provide any concrete information to back up that estimate.
--

	3. Page 12, lines 52-54: Did the authors analyze these COI statements to see what kind of conditions were set out for receiving funding? 4. Page 13, Limitations: Another limitation that should be mentioned is that the authors did not contact the associations directly to confirm the thoroughness of the website searches and therefore may have missed statements about funding.
--	---

REVIEWER	Martin McKee London School of Hygiene and Tropical Medicine, UK
REVIEW RETURNED	27-Jan-2019

GENERAL COMMENTS	This paper adds to a growing literature on the commercial determinants of health and conflicts of interest. Given the record of baby milk substitute manufacturers, it really is remarkable that any paediatric association would feel that it was appropriate to take money from them, but obviously they do. The limitation of this paper, which the authors note, is that it is at most a minimum estimate of the scale of the problem. They were unable to access online data on a substantial number of associations and, even for those that did, it seems quite likely that there would be funding that they were unable to identify. In fact, the challenges of tracing corporate funding are very substantial, as we showed recently with Coca-Cola (Serôdio PM et al. Public Health Nutr. 2018;21(9):1594-1607). That study only looked at funding of published research, and even that was extremely complicated. Consequently, I suspect that the authors will have missed quite a lot, especially funding of stands at conference exhibitions. That said, providing they stress that this is an extremely conservative estimate of the scale of what is happening, what they find is already quite alarming. I felt that the introduction could be more concise and to the point. It seemed that there were four key points. First, breastfeeding has important benefits for health. I don't think that they need all of the detail that is there, as BMJ readers will not need convincing. Second, and currently to some extent missing from what they have, I think they need to make clear that the industry has a dreadful record of employing practices that undermine efforts to promote breastfeeding. In essence, this is their opportunity to make the very strong case for not accepting funding. Third, their point that the WHO has called for safeguards against conflict of interest is important and could come here. Fourth, despite this, there is a lot of anecdotal evidence that there is a problem. These points can be made quite briefly. Turning to the methods, can they say anything about the results when the two research assistants compared their searches? Can they also say little bit more about the Baby Milk Action website, including its strengths and weaknesses? The results are fairly straightforward, given the limitations noted above. Turning to the discussion, is there any evidence that they can draw on that is specific to these products? Much of what they have is fairly generic, but without knowing the literature in detail, I would have thought that a call to Patti Rundall at Baby Milk Action would provide any material that exists. If, for example, they could illustrate the discussion with some of the case studies that have been published on, for example, the tobacco and soft drinks industries it would greatly strengthen their case for action.,
--

REVIEWER	Cristin Kearns University of California San Francisco
REVIEW RETURNED	30-Jan-2019

GENERAL COMMENTS	Introduction: For the reader unfamiliar with guidance on breastfeeding/infant formula, it would be helpful to have a clear description of the current guidelines, examples of inappropriate marketing by BMS manufacturers. How is the 2016 WHA resolution ending the inappropriate promotion of foods for infants and young children different from the International Code of Marketing of BMS? Why was it necessary for the WHA to create the new resolution? What are the anecdotal examples of a relationship between the BMS industry and pediatric associations? p7 Line 27: Citation is incorrect There also seems to be an important distinction between breastmilk substitutes, which are never appropriate to promote, and other foods for infants and children that are sometimes appropriate to promote. WHO, 2016b, paragraph 17, related to health professional associations, appears to be referring to other foods for infants and children, and not breastmilk substitutes per se. Therefore the WHA seems to be recommending that health professional associations avoid conflicts of interest with any company manufacturing any foods for infants and children, even if they meet the marketing requirements. Is this correct? Methods: The methods do not contain enough description to allow the study to be repeated. How were searches on the association websites conducted? Were search terms used? Was every page on the website read? How similar were the searches conducted by both research assistants on the 10 associations? Page 9, Line 16 implies all data collected were reviewed by both research assistants, did they both review all websites, or just the 10? Results: P10, Line 33 - this paragraph would be better supported if a table of BMS company websites was included. It would also provide the reader with an on overview of the companies involved. P10 - suggest switching order of last/2nd to last paragraph to maintain continuity with Table 2. P11 Line 3 - The IPA has a policy on conflicts of interest with the BMS industry in line with the WHO International Code on the Marketing of Breast Milk Substitutes and its subsequent biannual amendments of the World Health Assembly up until the IPA policy was published in 2014. Does the policy apply to IPA member associations? Also, because the IPA COI policy has not been updated since the 2016 WHA resolution, it does not include the guidelines on COI with companies manufacturing any infant foods and beverages. Do BMS companies also manufacture foods for infants and children? If so, isn't it possible that BMS company sponsorship documented in this paper (if the BMS companies were promoting acceptable foods and beverages and not breast milk substitutes) was in adherence with IPA policy as of 2014?
---

	Perhaps the study objective might better be stated as determining the extent to which national and regional paediatric associations adhere to the WHA 2016 resolution, now that it expands the scope of the WHO International Code on the Marketing of Breast Milk Substitutes to include all foods for infants and children (if I am interpreting the change correctly). Therefore, the results can serve as a baseline for implementation of the new recommendations.
--	---

VERSION 1 – AUTHOR RESPONSE

Reviewer: 1

Comment. This study is a systematic look at the issue of where medical societies get their funding and documents the extent to which paediatric associations receive funding from companies that make breast milk substitutes. It is important to highlight conflicts of interest between medical societies and commercial interests for the reasons that the authors outline.

Response. Thank you.

Comment. 1. Page 6, line 55: There should be a citation included to support the statement about the way that BMS companies are involved in health care.

Response. We have inserted 3 citations that document the magnitude of BMS marketing occurring in health care settings around the world.

Comment. 2. Page 12, lines 32-34: The authors should make it clear that the 30-50% figure is Dalsing's personal estimate and that he doesn't provide any concrete information to back up that estimate.

Response. We have reworded the sentence to be clearer that this was just Dalsing making a personal estimate.

Comment. 3. Page 12, lines 52-54: Did the authors analyze these COI statements to see what kind of conditions were set out for receiving funding?

Response. We have added a description of the COI statements at the end of the Results section.

Comment. 4. Page 13, Limitations: Another limitation that should be mentioned is that the authors did not contact the associations directly to confirm the thoroughness of the website searches and therefore may have missed statements about funding.

Response. This limitation has been added.

Reviewer: 2

Comment. This paper adds to a growing literature on the commercial determinants of health and conflicts of interest. Given the record of baby milk substitute manufacturers, it really is remarkable that any paediatric association would feel that it was appropriate to take money from them, but obviously they do.

The limitation of this paper, which the authors note, is that it is at most a minimum estimate of the scale of the problem. They were unable to access online data on a substantial number of associations and, even for those that did, it seems quite likely that there would be funding that they were unable to identify. In fact, the challenges of tracing corporate funding are very substantial, as we showed recently with Coca-Cola (Serôdio PM et al. Public Health Nutr. 2018;21(9):1594-1607). That study only looked at funding of published research, and even that was extremely complicated.

Consequently, I suspect that the authors will have missed quite a lot, especially funding of stands at

conference exhibitions. That said, providing they stress that this is an extremely conservative estimate of the scale of what is happening, what they find is already quite alarming.

Response. Thank you.

Comment. I felt that the introduction could be more concise and to the point. It seemed that there were four key points. First, breastfeeding has important benefits for health. I don't think that they need all of the detail that is there, as BMJ readers will not need convincing. Second, and currently to some extent missing from what they have, I think they need to make clear that the industry has a dreadful record of employing practices that undermine efforts to promote breastfeeding. In essence, this is their opportunity to make the very strong case for not accepting funding. Third, their point that the WHO has called for safeguards against conflict of interest is important and could come here. Fourth, despite this, there is a lot of anecdotal evidence that there is a problem. These points can be made quite briefly.

Response. We have removed some of the text from the introduction but added a few sentences on the activities of industry in aggressively promoting BMS. As requested by Reviewer 3, we have also added some examples of how industry uses health care workers to promote its products. The order of the introduction reflects the points listed above. We have not shortened the paragraph on breastfeeding benefits because, while readers likely understand the importance of breastfeeding, the magnitude of the problem is probably less well-recognized.

Comment. Turning to the methods, can they say anything about the results when the two research assistants compared their searches? Can they also say little bit more about the Baby Milk Action website, including its strengths and weaknesses?

Response. We have added a sentence clarifying the results of the cross-over analysis. Regarding the Baby Milk Action website, it is out of the scope of this paper to evaluate this website. We simply used the website to provide a list of BMS companies for further searching.

Comment. The results are fairly straightforward, given the limitations noted above.

Response. Thank you.

Comment. Turning to the discussion, is there any evidence that they can draw on that is specific to these products? Much of what they have is fairly generic, but without knowing the literature in detail, I would have thought that a call to Patti Rundall at Baby Milk Action would provide any material that exists. If, for example, they could illustrate the discussion with some of the case studies that have been published on, for example, the tobacco and soft drinks industries it would greatly strengthen their case for action.,

Response. We have added an example that was published in BMJ last year about a dramatic rise in prescriptions for specialized formula for cow's milk protein allergy. There appear to have been significant conflicts of interest on the expert panels that have published guidelines on the topic.

Reviewer: 3

Comment. Introduction: For the reader unfamiliar with guidance on breastfeeding/infant formula, it would be helpful to have a clear description of the current guidelines, examples of inappropriate marketing by BMS manufacturers. How is the 2016 WHA resolution ending the inappropriate promotion of foods for infants and young children different from the International Code of Marketing of BMS? Why was it necessary for the WHA to create the new resolution? What are the anecdotal examples of a relationship between the BMS industry and pediatric associations?

Response. We have added a paragraph in the introduction giving examples of inappropriate marketing through health workers and facilities. However, a broader description of the 2016 WHA resolution is beyond the scope of this paper. While the resolution adds to the rationale for ending BMS company sponsorship of pediatric associations, the issue would still be important even without the resolution. Reviewer 1 asked for a significantly shorter introduction, so we have opted not to add

this text and instead simply insert an additional reference to a fuller description of the Guidance on ending inappropriate promotion of foods for infants and young children.

Comment. p7 Line 27: Citation is incorrect
Response. Corrected.

Comment. There also seems to be an important distinction between breastmilk substitutes, which are never appropriate to promote, and other foods for infants and children that are sometimes appropriate to promote. WHO, 2016b, paragraph 17, related to health professional associations, appears to be referring to other foods for infants and children, and not breastmilk substitutes per se. Therefore the WHA seems to be recommending that health professional associations avoid conflicts of interest with any company manufacturing any foods for infants and children, even if they meet the marketing requirements. Is this correct?

Response. This is a correct interpretation. The recommendation to avoid conflicts of interest applies to all foods and beverages targeted for infants and young children. This paper is focused on BMS and we prefer not to confuse the reader by going into the somewhat more complicated issues on the marketing of complementary foods.

Comment. Methods: The methods do not contain enough description to allow the study to be repeated. How were searches on the association websites conducted? Were search terms used? Was every page on the website read? How similar were the searches conducted by both research assistants on the 10 associations? Page 9, Line 16 implies all data collected were reviewed by both research assistants, did they both review all websites, or just the 10?

Response. We have added a paragraph to describe more about the search within each website. Further information on the results of the cross-over reviewing has been added. While the actual review of the websites were done separately by the two reviewers, the checking of the spreadsheet was done by each of them to ensure information was consistently documented. We believe that this distinction is clear in the text.

Comment. Results: P10, Line 33 - this paragraph would be better supported if a table of BMS company websites was included. It would also provide the reader with an on overview of the companies involved.

Response. Because the review of company websites provided such limited information about funding and sponsorship (only 3 additional funding instances identified and purpose of funding not documented), we do not believe that adding this information would strengthen the paper.

Comment. P10 - suggest switching order of last/2nd to last paragraph to maintain continuity with Table 2.

Response. Done.

Comment. P11 Line 3 - The IPA has a policy on conflicts of interest with the BMS industry in line with the WHO International Code on the Marketing of Breast Milk Substitutes and its subsequent biannual amendments of the World Health Assembly up until the IPA policy was published in 2014. Does the policy apply to IPA member associations?

Also, because the IPA COI policy has not been updated since the 2016 WHA resolution, it does not include the guidelines on COI with companies manufacturing any infant foods and beverages. Do BMS companies also manufacture foods for infants and children? If so, isn't it possible that BMS company sponsorship documented in this paper (if the BMS companies were promoting acceptable foods and beverages and not breast milk substitutes) was in adherence with IPA policy as of 2014?

Response. The IPA policy does not apply to member associations and thus it would not be relevant here. Each association establishes its own funding policies, as is evident by the wide variety of policies and practices observed in this study.

Comment. Perhaps the study objective might better be stated as determining the extent to which national and regional paediatric associations adhere to the WHA 2016 resolution, now that it expands the scope of the WHO International Code on the Marketing of Breast Milk Substitutes to include all foods for infants and children (if I am interpreting the change correctly). Therefore, the results can serve as a baseline for implementation of the new recommendations.

Response. While the WHA 2016 resolution clearly spelled out a recommendation to end sponsorship of professional associations, this is not the only reason to call for it. The issues of conflict of interest have been raised for many years and earlier resolutions have called for avoidance of conflicts of interest with regard to health care professionals and BMS companies. Thus, we believe that the objective of documenting the extent of BMS sponsorship of national and regional paediatric associations is appropriate, without specific reference to WHA 69.9.

VERSION 2 – REVIEW

REVIEWER	Joel Lexchin York University Canada In 2016-2019, Joel Lexchin was a paid consultant on two projects: one looking at developing principles for conservative diagnosis (Gordon and Betty Moore Foundation) and a second deciding what drugs should be provided free of charge by general practitioners (Government of Canada, Ontario Supporting Patient Oriented Research Support Unit and the St Michael's Hospital Foundation). He also received payment for being on a panel at the American Diabetes Association, for a talk at the Toronto Reference Library, for writing a brief for a law firm and from the Canadian Institutes of Health Research for presenting at a workshop on conflict-of-interest in clinical practice guidelines. He is currently a member of research groups that are receiving money from the Canadian Institutes of Health Research and the Australian National Health and Medical Research Council. He is member of the Foundation Board of Health Action International and the Board of Canadian Doctors for Medicare. He receives royalties from University of Toronto Press and James Lorimer & Co. Ltd. for books he has written.
REVIEW RETURNED	05-Apr-2019
GENERAL COMMENTS	All of my initial comments have been addressed.
REVIEWER	Martin McKee LSHTM, UK
REVIEW RETURNED	09-Apr-2019
GENERAL COMMENTS	The issues I raised have been addressed
REVIEWER	Cristin Kearns, DDS, MBA University of California San Francisco, United States
REVIEW RETURNED	19-Apr-2019
GENERAL COMMENTS	The authors have done a nice job addressing previous comments. I have no additional comments.